# Hurdle Technology Approach to Control *Listeria monocytogenes* Using Rhamnolipid Biosurfactant

**DOI:** 10.3390/foods12030570

**Published:** 2023-01-28

**Authors:** Lowieze Lenaerts, Tathiane Ferroni Passos, Elisa Gayán, Chris W. Michiels, Marcia Nitschke

**Affiliations:** 1Department of Microbial and Molecular Systems, Leuven Food Science and Nutrition Research Center (LFoRCe), KU Leuven, Kasteelpark Arenberg 22, B-3001 Leuven, Belgium; 2São Carlos Institute of Chemistry (IQSC), University of São Paulo, Trabalhador São-Carlense Av., 400, P.O. Box 780, São Carlos 13560-970, São Paulo, Brazil; 3Department of Animal Production and Food Science, AgriFood Institute of Aragon (IA2), Faculty of Veterinary, University of Zaragoza-CITA, Miguel Servet 177, 50013 Zaragoza, Spain

**Keywords:** antimicrobial activity, biosurfactant, hurdle technology, *Listeria monocytogenes*, rhamnolipid, low moisture foods

## Abstract

This study evaluates the combination of mild heat with a natural surfactant for the inactivation of *L. monocytogenes* Scott A in low-water-activity (a_w_) model systems. Glycerol or NaCl was used to reduce the a_w_ to 0.92, and different concentrations of rhamnolipid (RL) biosurfactant were added before heat treatment (60 °C, 5 min). Using glycerol, RL treatment (50–250 µg/mL) reduced bacterial population by less than 0.2 log and heat treatment up to 1.5 log, while the combination of both hurdles reached around 5.0 log reduction. In the NaCl medium, RL treatment displayed higher inactivation than in the glycerol medium at the same a_w_ level and a larger synergistic lethal effect when combined with heat, achieving ≥ 6.0 log reduction at 10–250 µg/mL RL concentrations. The growth inhibition activity of RL was enhanced by the presence of the monovalent salts NaCl and KCl, reducing MIC values from >2500 µg/mL (without salt) to 39 µg/mL (with 7.5% salt). The enhanced antimicrobial activity of RL promoted by the presence of salts was shown to be pH-dependent and more effective under neutral conditions. Overall, results demonstrate that RL can be exploited to design novel strategies based on hurdle approaches aiming to control *L. monocytogenes*.

## 1. Introduction

Heat processing is the most extensively utilized commercial food preservation method to destroy microorganisms and thereby extend the shelf life and ensure the microbiological safety of food. However, intensive heat treatments result in undesirable nutritional and organoleptic changes in the final product [1,2]. The tendency for more natural and healthy foods has prompted the development of mild processing methods that can replace traditional heat treatments or reduce their intensity while retaining the nutritional and sensory properties and safety of products [3,4]. A reduction of treatment intensity can be achieved when heat is combined with other preservation methods that together ensure the appropriate inactivation of pathogenic and spoilage microorganisms. Such combined treatments can also be used to enhance treatment efficiency when intrinsic food properties, such as low water activity (a_w_), tend to protect bacteria against heat [5]. Within this context, hurdle approaches resulting in a synergistic lethal effect are desired in order to significantly reduce the intensity of each individual preservation method while obtaining foods with better quality and lower processing cost [6].

On the other hand, the food industry is making increasing efforts to replace chemical food preservatives by natural alternatives [7]. Although some natural compounds, such as those from plant essential oils, have a strong antimicrobial effect, they often have low solubility and a strong impact on the sensorial properties of foods [8,9]. Therefore, hurdle approaches wherein natural antimicrobials at lower concentrations are combined with mild processing are being explored to reduce their negative impact on food products.

Microbial surfactants or biosurfactants (BS) are a natural class of surface-active compounds produced by microorganisms. The possibility of producing BS from renewable feedstocks or agricultural wastes, their biodegradability and low toxicity align well with the concept of “green chemistry”, which represents an important tool for innovation and sustainability, fulfilling actual market needs [10]. The rhamnolipids (RLs) produced by *Pseudomonas aeruginosa* are glycolipid biosurfactants with several useful characteristics, and they are considered promising multipurpose ingredients in food processing [11]. Rhamnolipids are approved by the US Environmental Protection Agency (EPA) as antimicrobials for the washing or spraying of agricultural crops and are classified by the EPA in the lowest toxicity category (IV), having an acute oral LD_50_ greater than 5000 mg/kg [12].

*Listeria monocytogenes* are widespread in diverse natural environments such as soil, water, plants and the gastro-intestinal tract of humans and animals [13]. They can also establish and thrive in many food production environments, and their presence and persistence in the food supply chain is a matter of concern, as they are the causative agent of a serious foodborne illness known as listeriosis [14]. The consumption of contaminated foods, including chilled meat, poultry, seafood, vegetables, fruits, dairy and ready-to-eat products, is responsible for outbreaks worldwide [15]. A factor contributing to the success of the pathogen is its ability to grow over a wide range of pHs (4.5–9.5), salt concentrations (up to 10%) and a_w_ (down to 0.92) [16,17]. In addition, it can form biofilms and resist sanitizers [18], increasing its ability to contaminate food products. Finally, the pathogen has a remarkable aptitude for long-term survival under stressful conditions. For example, *L. monocytogenes* ATCC 19115 was reported to survive in culture medium at pH 4.0 and 4°C for 19 days in the presence of 21% NaCl [19].

In a previous study, it was demonstrated that a purified RL produced by *P. aeruginosa* PA1 inhibited the growth of *L. monocytogenes* ATCC 19112 and ATCC 7644 by 100% [20]. Further work, conducted with 32 *L. monocytogenes* isolates, revealed that over 90% were susceptible to RL with MIC values ranging from 78.1 to 2500 µg/mL. The study also demonstrated that RL had a synergistic effect when combined with nisin, probably because both molecules act on the same cell target [21]. The exact mechanism behind RLs’ antimicrobial activity remains unclear, but most researchers anticipate that they disturb the cell membrane due to their amphipathic nature, leading to increased permeability, metabolite leakage and eventually cellular lysis [21,22].

It is well known that reducing microbial contamination of low a_w_ foods by thermal processing requires very intensive treatment conditions due to the increased heat tolerance of microorganisms [23,24]. Such treatments therefore cause heat damage, leading to unacceptable quality deterioration. Several membrane-targeting antimicrobials show a synergistic lethal effect when combined with mild heat treatment [25,26,27]; however, the effectiveness of synergistic combinations has hardly been studied in low-moisture systems. Therefore, in this work, we investigated whether the combination of RL and mild heat treatment is effective, and possibly synergistic, in inactivating *L. monocytogenes* in low-moisture systems. Both a nonionic (glycerol) and ionic (NaCl) solute were used for reducing the a_w_ of a model system, and the effect of RL, heat and their combination was evaluated. Further, the influence of ionic strength on the growth-inhibitory activity of RL is also studied and discussed.

## 2. Materials and Methods

### 2.1. Biosurfactant

Commercial rhamnolipid solution (99% purity) was provided by Rhamnolipid Inc. (Tampa, FL, USA), containing approximately 54.4% RhaC_10_C_10_ and 24.2% Rha_2_C_10_C_10_ as major components.

### 2.2. Bacterial Strain and Culture Conditions

*L. monocytogenes* Scott A, acquired from the International Life Sciences Institute (ILSI), North America [28], was used throughout this investigation. The strain was maintained at −80 °C in Brain Heart Infusion (BHI; Oxoid, Basingstoke, UK) broth supplemented with 25% glycerol. For cell revitalization, a loopful of frozen culture was spread on BHI agar and incubated for 24 h at 30 °C. Stationary-phase cultures were prepared by inoculating a test tube containing 4 mL of BHI broth with a single colony from the stock plate and incubating it aerobically in an orbital shaker at 200 rpm (GFL 3005, Burgwedel, Germany) for 19 ± 1 h at 30 °C.

### 2.3. Inactivation of L. monocytogenes by Heat and RL in Reduced Moisture Systems

A 4 mL stationary phase culture of *L. monocytogenes* Scott A was transferred to a 10 mL tube and centrifuged at 3829× *g* for 5 min. After the removal of the supernatant, the cells were washed twice with 4 mL of 10 mM sodium phosphate buffer (pH 6.7) and centrifuged again. The pellet was finally suspended in 7.5 mL of buffer with the appropriate solutes to adjust the cell concentration to ~1.3 × 10^9^ CFU/mL.

The inactivation assays were conducted in 10 mM sodium phosphate buffer (pH 6.7) adjusted to a_w_ 0.98 or 0.92 with glycerol (6.95% and 29.63% (*w*/*w*) for a_w_ 0.98 and 0.92, respectively) or NaCl (3.63% and 14.94% (*w*/*w*) for a_w_ 0.98 and 0.92, respectively) as model systems of low-moisture matrices. The water activity was measured using a HygroPalm AW1 hygrometer (Rotronics AG, Bassersdorf, Germany). When necessary, the buffer was supplemented with the appropriate RL concentration (1 to 250 µg/mL). For RL treatment at room temperature, 100 μL of the cell suspension (~1.3 × 10^9^ CFU/mL) was added to an Eppendorf tube containing 900 µL of the buffer with RL added and incubated for 5 min at ~20 °C. For heat treatment alone or in combination with RL, the same volume of cell suspension was added to an Eppendorf tube containing 900 µL of the buffer without or with RL that was previously prewarmed at 60 °C in a heating block (VWR, Radnor, PA, USA), and then the samples were incubated at the same temperature for 5 min. As such, the initial cell concentration for all RL, heat and the combined RL and heat treatment was ~1.3 × 10^8^ CFU/mL. After treatment, the samples were serially diluted, and the number of viable cells was determined by the drop plate method [29] (5 μL drops of each dilution) on BHI agar. Plates were incubated at 30 °C for 24 h, and then spots containing 5–50 colonies were counted, so that the quantification limit was 1000 CFU/mL. The logarithmic reduction factor (LRF, dimensionless unit) was calculated as log(*N*_0_/*N*), where *N*_0_ and *N* are the viable cell concentration (CFU/mL) at the start and at the end of the treatment, respectively.

To determine the occurrence of synergistic/additive interactions between the lethal effect of heat and RL, the LRF of the combined treatment was statistically compared to the sum of the LRFs of the individual treatments (theoretical additive inactivation), as described by Feyaerts et al. [30].

### 2.4. Growth Inhibitory Activity

The minimum inhibitory concentration (MIC) and minimum bactericidal concentration (MBC) of the RL were determined using the micro-broth dilution technique according to the Clinical and Laboratory Standards Institute guidelines [31], using concentrations from 2500.0 to 4.9 µg/mL. The MBC was determined by transferring 100 µL from the wells of the MIC microtiter plates where no growth was observed (after 24 h of incubation) to the surface of BHI agar plates that were subsequently incubated for 48 h at 30 °C. The MBC was considered the lowest concentration of RL at which no colonies were formed [32].

### 2.5. Growth Curves

Growth experiments were conducted in BHI medium in 96-well microplates covered with adhesive foil in an automated microplate reader (Multiskan Ascent, Thermo Fisher Scientific, Waltham, MA, USA) at 30 °C [33]. An aliquot of 20 µL of an overnight culture diluted to a cell population of 10^7^ CFU/mL was added to 180 µL of BHI supplemented with 500 µg/mL of RL, and the optical density (OD, 620 nm) was measured at defined time intervals. To study the influence of NaCl or KCl on the antimicrobial activity of RL, the salts were added to the culture medium at concentrations ranging from 1.0% to 7.5% (*w*/*v*).

### 2.6. Microscopy

Fluorescence microscopy was performed with a Ti-Eclipse inverted microscope (Nikon, Tokyo, Japan) after cell staining with a LIVE/DEAD BacLight Bacterial Viability kit (Thermo Fisher Scientific) following supplier specifications [34]. Images were acquired and processed using NIS-Elements (Nikon, Tokyo, Japan) and visualized using the open software ImageJ (https://imagej.net/ij/index.html, version downloaded on 1 March 2019).

### 2.7. CMC Determination

The critical micelle concentration (CMC) was determined by surface tension measurements of surfactant dilutions using the Du Nouy ring method on a Sigma700/701 tensiometer (Attention, Helsinki, Finland), and calculated by the equipment software. The measurements were done in distilled water with different concentrations of NaCl and KCl as required. The pH of such solutions was adjusted using HCl or NaOH (0.1 M) when needed.

### 2.8. Statistics

Inactivation and growth curve data were expressed as the mean of at least three independent replicate experiments performed on different days. MIC values were expressed as the mode of at least five independent replicates. One-way analysis of variance (ANOVA) and *t*-tests were performed with the software JMP (Version 14; SAS Institute Inc., Cary, NC, USA). For the comparison of individual treatments and the determination of synergistic/additive lethal effects, differences were considered significant when *p* ≤ 0.05.

## 3. Results and Discussion

### 3.1. Effect of RL on Heat Inactivation at Low Water Activity

Figure 1 shows the heat inactivation (60 °C, 5 min) of *L. monocytogenes* Scott A in sodium phosphate buffer (a_w_ > 0.99) and in the same buffer with reduced a_w_ from >0.99 to 0.98 or 0.92 by the addition of the solutes. While reducing a_w_ to 0.98 by either glycerol or NaCl did not significantly (*p* > 0.05) change inactivation, further reduction to a_w_ 0.92 exerted a strong protective effect, decreasing inactivation by 3.5 to 4.3 log cycles compared to the control (a_w_ > 0.99).

It is well known that bacterial heat resistance increases when reducing the a_w_ of the treatment medium, as has been documented for *L. monocytogenes* [35]. This phenomenon is attributed to the reduced mobility of water molecules and the increased stability of proteins and ribosomes promoted by the presence of solutes, although this effect is dependent on the type of solute, treatment temperature and bacterial strain [23,24]. Due to the large protective effect against heat observed at a_w_ 0.92, further experiments were conducted at this condition.

Subsequently, we studied the inactivation of *L. monocytogenes* Scott A by heat (60 °C, 5 min) combined with RL at a_w_ 0.92 in comparison with the inactivation of each individual treatment. To verify whether RL lethality had a concentration-dependent effect, a range of concentrations from 1 to 250 µg/mL was studied. Using glycerol as a solute, less than 0.2 log cycles of inactivation was observed when RL was added, whereas the heat treatment (without RL) caused about 1.5 log reduction (Figure 2A). Notably, the combination of both hurdles considerably boosted the inactivation to around 5.0 log cycles for RL concentrations from 50 µg/mL to 250 µg/mL, indicating the existence of a synergistic lethal effect. At lower RL concentrations, the degree of synergy between RL and heat progressively decreased until it disappeared at 1 µg/mL.

Using NaCl as a solute, RL displayed a concentration-dependent bactericidal effect at room temperature, with inactivation ranging from 0.4 log cycles at 10 µg/mL to 1.8 log cycles at 250 µg/mL (Figure 2B). The synergy of RL with heat was also stronger in the presence of NaCl than of glycerol as a solute: *L. monocytogenes* population was reduced by ≥6.0 log cycles at RL concentrations between 25 µg/mL and 250 µg/mL. The above results suggest that RL may offer potential to make the heat treatment of foods with reduced a_w_ more effective.

There is no data available regarding microbial inactivation by the combination of heat and RL; however, there exist similar studies using other natural antimicrobials in high-moisture matrices. For instance, the population of *L. monocytogenes* Scott A present in semi-skimmed milk was reduced by 4 log cycles after combining heat (60 °C, 6 min) with vanillin (1400 ppm) [36]. The combination of essential oils from *Laurus mobilis* and *Myrtus communis L.* (0.2 µL/mL) with mild heat (54 °C, 10 min) in buffer systems (pH 7.0) resulted in the synergistic inactivation of *L. monocytogenes* EGD-e, reducing the population by 3.5 log cycles [37]. The thermal resistance of *L. monocytogenes* STCC 4032 was reduced by 1–4 log cycles by 0.5 mM D-limonene at 55 °C (1 min), depending on whether the compound was added directly or in the form of a nanoemulsion [38].

It has been demonstrated that the lethality of natural antimicrobials may be reduced in low-water-activity systems, probably due to lower chances of contact between the compounds and the cell target [39,40]. As such, the inactivation of *Salmonella* Tennessee by cinnamaldehyde (125–500 mg/L), carvacrol (125–500 mg/L) or lauric arginate (50–200 mg/L) after 3 days of incubation (25 °C) was dramatically reduced when decreasing a_w_ from >0.99 to 0.7–0.3 in a glycerol–sucrose model and peanut paste [40]. However, we found only one study on the effectiveness of synergistic heat and antimicrobial combinations in low-moisture systems, in which it was reported that the synergetic lethal effect between heat and oregano essential oil (1–2%) or Ԑ-polylysine (0.4%) in diluted tahini (a_w_ > 0.99) was completely lost in undiluted tahini (a_w_ 0.256–0.335) [39]. The reason for this effect may be that the antimicrobials are concentrated in the oil phase in undiluted tahini. Our experiments were conducted in a fat-free medium and showed a strong synergy between RL and heat at a_w_ 0.92, suggesting that such a hurdle approach could be effective for processing low-fat low-water-activity foods.

The antimicrobial action of RL is based on disturbance of the cell membrane integrity [41], although the precise details of the mechanism are unknown. The generation of reactive oxygen species (ROS) was also proposed to contribute to the bactericidal effect of rhamnolipids [42]. As a surfactant, RL molecules can self-assemble into structures such as micelles, vesicles and lamellas [43] at concentrations above their critical micelle concentration (CMC). Actually, the role of RL aggregation in its antimicrobial activity is not well understood: while surfactant monomers can easily insert themselves into the cell membrane, the presence of molecular aggregates can enhance local surfactant concentration and charge [44]. According to Rodrigues et al. [45], the antimicrobial activity of RL against two *Aspergillus* species was associated with increasing micelle size and with the presence of vesicle-like molecular aggregates. Factors, such as temperature, pH, ionic strength and surfactant concentration, influence the type, size and charge of molecular self-assembly structures [46,47,48], and therefore, it is reasonable to assume that changes in these parameters will have an impact on RL antimicrobial activity. Temperature influences CMC, and such an effect is dependent on the type of surfactant. For ionic surfactants, such as RL, an increase in temperature can reduce the hydration of the hydrophilic head group favoring the formation of micelles and consequently decreasing their CMC [49].

It is important to note that the combined treatment was more effective with NaCl as a solute than with glycerol at the same a_w_ and at all RL concentrations tested (Figure 2). These results suggest that, although heat increases the antimicrobial effect of RL in the presence of both solutes, NaCl and glycerol interact differently with RL. Possibly, the ionic nature of NaCl may enhance RL antimicrobial activity compared to glycerol. Given the strong lethal effect of RL in the presence of NaCl, we proceeded to investigate whether the combination of RL and NaCl exerted a growth inhibitory effect on *L. monocytogenes*.

### 3.2. Growth Inhibitory Activity of RL in Combination with Salts

The growth kinetics of *L. monocytogenes* in BHI with 0.0%, 1.0%, 2.0%, 5.0% and 7.5% NaCl with and without 500 µg/mL of RL is shown in Figure 3. In the absence of RL, lag phases increased and growth rates decreased especially at 7.5% NaCl but not at lower concentrations (Figure 3A). The addition of RL (500 µg/mL) to the medium impaired bacterial growth but also rendered the bacteria more sensitive to NaCl, even at the lowest concentration used (1%) (Figure 3B). At the highest NaCl concentrations (5.0–7.5%), growth was completely inhibited. The viability staining of the bacterial cells after growing in culture medium with 5% NaCl added (control), RL (500 μg/mL) and the combination of both (RL+NaCl) is displayed in Figure 4. Green viable cells were present in control medium and also after the treatment with RL for 4 h (Figure 4A) and 24 h (Figure 4B), whereas no live cells were visible after the combined treatment even after only 4 h. Thus, it can be concluded that NaCl and RL inactivate *L. monocytogenes* in a synergistic manner.

While NaCl is one of the most commonly used food additives, and many foods depend partly on NaCl for their microbiological stability and safety, the growing pressure for the reduction of dietary sodium intake is pushing food producers to replace NaCl by substitutes such potassium salts [50]. Therefore, we evaluated if KCl and RL (500 µg/mL) also synergistically inhibited the growth of *L. monocytogenes*. Remarkably, although KCl was slightly less inhibitory than NaCl when used alone, it had a stronger synergistic effect when combined with RL than NaCl (Figure 5). Even at the lowest KCl concentration (1.0%), growth was completely inhibited in combination with RL (500 µg/mL).

The interaction between RL and NaCl or KCl was investigated in more detail by determining the MIC and MBC values of RL at different salt concentrations (Table 1). In the absence of salts, no MIC or MBC value could be determined because growth was not completely inhibited even at the highest RL concentration. At salt concentrations of 1.0% and 2.0%, the MIC and MBC values of RL were considerably lower for KCl than for NaCl, while at 5.0% and 7.5%, there was no difference. These results confirm that KCl has a stronger synergistic effect than NaCl in combination with RL.

At the neutral pH of the BHI culture medium, the RL carboxyl groups are dissociated, resulting in a negatively charged structure [51], in which strong repulsive forces exist between the head groups of RL molecules. Cations of opposite charge (Na^+^ or K^+^) shield these negative charges and thus decrease the repulsive forces and lower the CMC [48,52]. Moreover, the addition of salts increases the micelle size and aggregation number of ionic surfactants [53]. Thus, micelles are formed at lower surfactant concentration, and more surfactant molecules are present in the micelles.

Na^+^ and K^+^ cations may have a different effect because of their different charge/size ratio. In water, the radius of a hydrated Na^+^ cation is larger than that of a hydrated K^+^ cation [50], and consequently, the binding of Na^+^ with surfactant molecules is weaker than that of K^+^. Sood and Aggarwal [53] reported that the CMC of the anionic surfactant sodium dodecylbenzene sulphonate was lower in the presence of KCl than in the presence of NaCl. In line with this behavior, the CMC of RL was lower in KCl than in NaCl considering the same molar concentration of both salts (Table 2). However, our antimicrobial assays were conducted using equal percentage (*w/v*) concentrations of salts (at pH 7.0), and, under such conditions, the CMC of RL was lower in NaCl than in KCl (Table 2). Thus, the enhanced antimicrobial activity of the combination of KCl and RL cannot be only attributed to decreased CMC, and therefore other specific interactions of K^+^ cations with cell surface components or in surfactant micelle organization may account for the observed results.

The carboxyl groups of RL molecules become predominantly protonated (uncharged) when the pH is lower than the pK_a_, which is 5.6 and 5.9 for di and mono-RL forms, respectively [43,54]. In previous work, it was observed that the antimicrobial activity of RL increased under acidic conditions [55], and authors postulated that the electrostatic repulsion between RL and the cell surface was reduced, favoring cell–surfactant interaction.

If the increased antimicrobial activity of RL at low pH and in the presence of salts were indeed due to the reduction or shielding of negative charges, it would be predicted that the cations would have little or no effect under acidic conditions wherein RL are nonionic. To test this hypothesis, we conducted a new set of experiments in BHI medium adjusted to pH 5.0. Figure 6 shows that *L. monocytogenes* Scott A was able to grow in BHI adjusted to pH 5.0 when 5% NaCl was added (control). Nevertheless, when the culture broth was supplemented with RL (500 µg/mL), the growth was inhibited independently of the presence of NaCl. Furthermore, the MIC value of RL at pH 5.0 was 19.5 µg/mL for both NaCl-supplemented and -non-supplemented media (data not shown). Comparatively, when RL were utilized under neutral pH (Table 1), the MIC was reduced from >2500 (without NaCl) to 78 µg/mL (with 5% NaCl). These results suggest that the nonionic form is more effective against *L. monocytogenes* and confirm that the effect of salts is observed only at pH values wherein negatively charged RL molecules predominate in solution. As such, they lend support to the hypothesis that the shielding of negative charges by monovalent cations makes RL behave as nonionic surfactants favoring their interaction with cells.

Table 2 shows that the CMC of RL decreased as NaCl concentration increased in solution. The addition of 5% NaCl lowered the CMC from 94.2 to 16.0 at pH 7.0 and the MIC from >2500 to 78 µg/mL (Table 1). The effect of salt on surfactant CMC was less evident at pH 5.0, probably due to the predominance of the nonionic form of the RL as discussed above (Table 2). The increase in pH also increased the CMC of the surfactant in the absence of salt and at specific ionic strength; however, this was not observed at the highest salt concentration. Our findings are in agreement with similar reports in literature regarding the CMC behavior of RL [56,57].

An interesting question is whether the increased antimicrobial effect observed when the negative charge of RL is reduced or shielded is only due to the decreased repulsion of RL and the bacterial surface or is also affected by their molecular aggregation. The morphology of RL molecular aggregates is also governed by pH. An increase of pH from 5.0 to 7.0 induced a shift of RL self-assembly structures from vesicles to lamellas, lipid particles and micelles [43,56]. It is known that membrane vesicles and micelles can fuse with bacterial membranes under particular conditions, and it is conceivable that such fusion events can strongly enhance the antimicrobial activity of surfactants. However, further work is needed to investigate the formation of such structures in relation to environmental conditions (pH, salts) and their interaction with bacterial cells.

## 4. Conclusions

A hurdle technology approach combining a rhamnolipid biosurfactant and mild heat revealed a strongly synergistic effect against *L. monocytogenes* in a low-water-activity model using ionic and sugar solutes. Furthermore, the growth-inhibitory activity of RL was also strongly enhanced by the presence of NaCl and KCl, suggesting that the overall antilisterial properties of rhamnolipids depend on the ionic strength of the medium and the surfactant CMC. The results obtained in this work contribute to the understanding of RL antimicrobial action and may open new perspectives regarding their applications in food processing. Notably, further research exploring the efficacy of this natural bio-based surfactant in combination with salts can result in innovative and sustainable strategies to control *L. monocytogenes* in the food chain.

## Figures and Tables

**Figure 1 foods-12-00570-f001:**
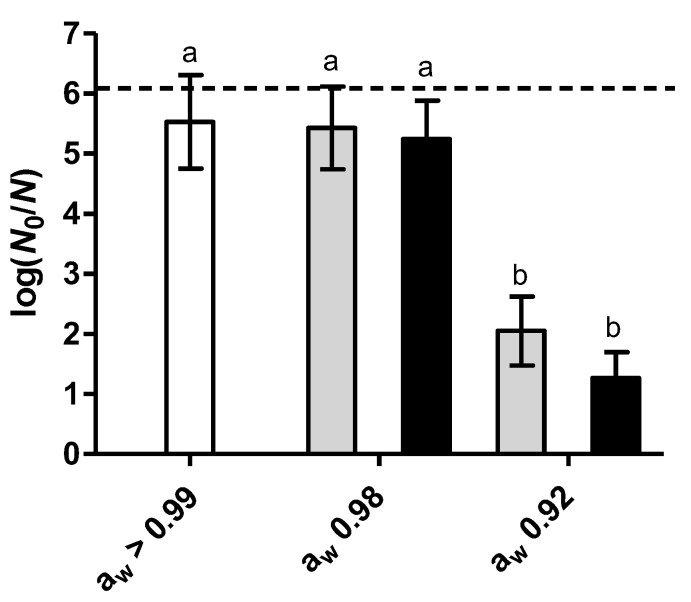
Logarithmic reduction factor (LRF; log(*N*_0_/*N*)) of *L. monocytogenes* Scott A by heat treatment (60 °C, 5 min) in sodium phosphate buffer (pH 6.7) at a_w_ > 0.99 (control, white bars), 0.98 or 0.92, adjusted with glycerol (gray bars) or NaCl (black bars). Error bars represent the standard deviation of the means of triplicates. Different letters indicate statistically significant differences (*p* ≤ 0.05) in the inactivation at different a_w_ levels and using different solutes. The dashed line indicates the LRF corresponding to the quantification limit of surviving cells (1000 CFU/mL).

**Figure 2 foods-12-00570-f002:**
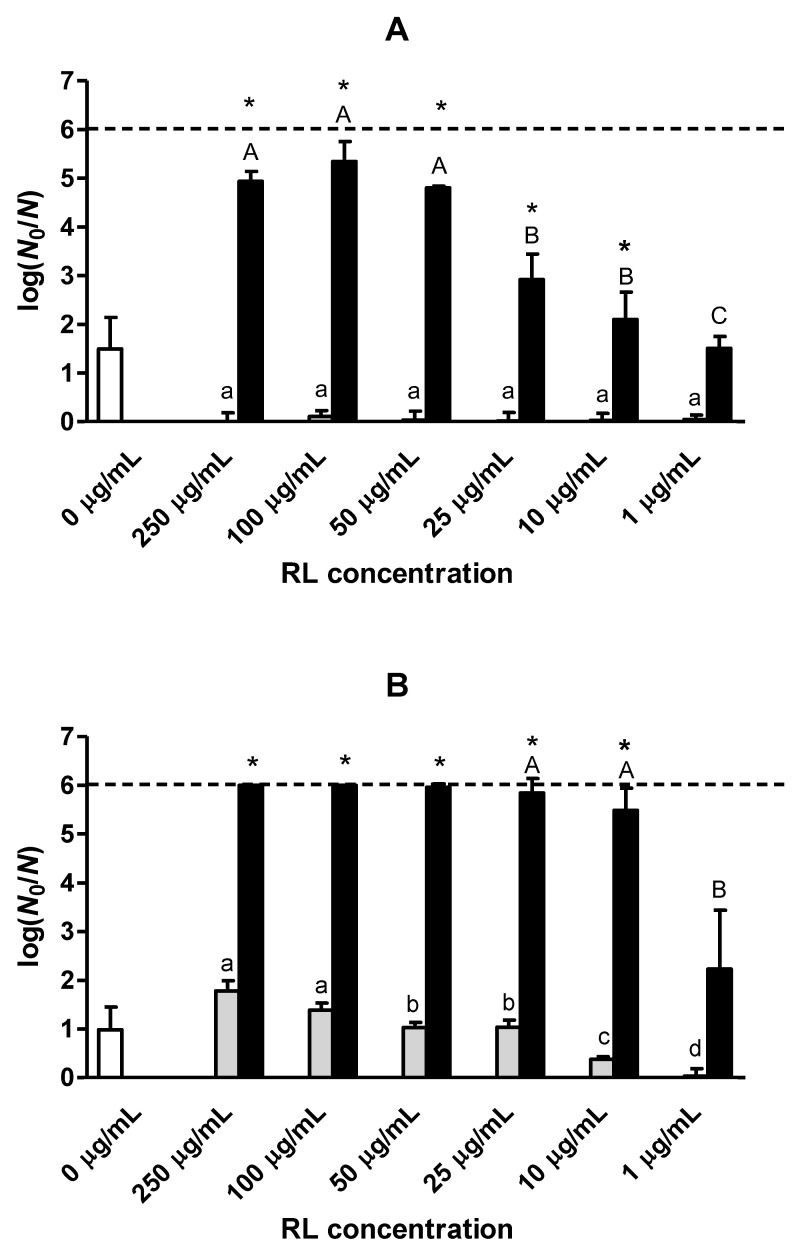
Logarithmic reduction factor (LRF; log(*N*_0_/N)) of *L. monocytogenes* Scott A in sodium phosphate buffer (pH 6.7) of a_w_ 0.92 with (**A**) glycerol or (**B**) NaCl by heat, RL at different concentrations and the combination of heat and RL. The white bar represents heat inactivation alone (60 °C, 5 min; without RL added). For each RL concentration, the gray bar presents inactivation by RL at room temperature (~20 °C, 5 min), while the black bar indicates inactivation by the combination of RL and heat (60 °C, 5 min). The dashed line indicates the maximum detectable LRF corresponding to the quantification limit (1000 CFU/mL). Error bars represent the standard deviations of the means of triplicates. Asterisk indicates statistically significant differences (*p* ≤ 0.05) between the theoretical additive inactivation by heat and RL and the experimental inactivation obtained by the combined treatment, and therefore the occurrence of a synergistic lethal effect. Lowercase and capital letters indicate statistically significant differences (*p* ≤ 0.05) in the inactivation by RL and by the combination of heat and RL treatment, respectively, among the different RL concentrations tested.

**Figure 3 foods-12-00570-f003:**
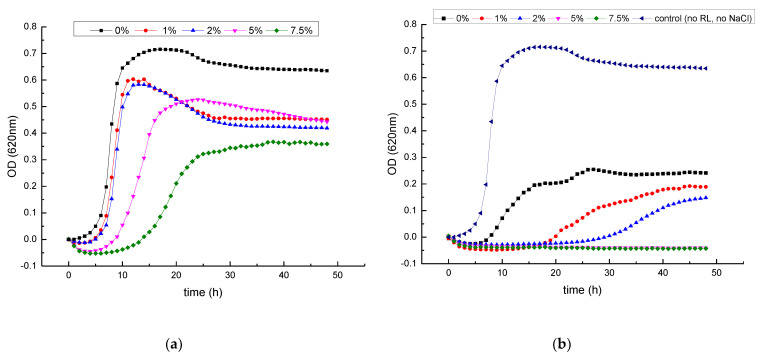
Growth kinetics of *L. monocytogenes* Scott A in BHI medium supplemented with (**a**) different concentrations of NaCl, without RL and (**b**) in combination with 500 µg/mL of RL at 30 °C. Data points represent the mean values of three replicates. Error bars have been omitted for clarity.

**Figure 4 foods-12-00570-f004:**
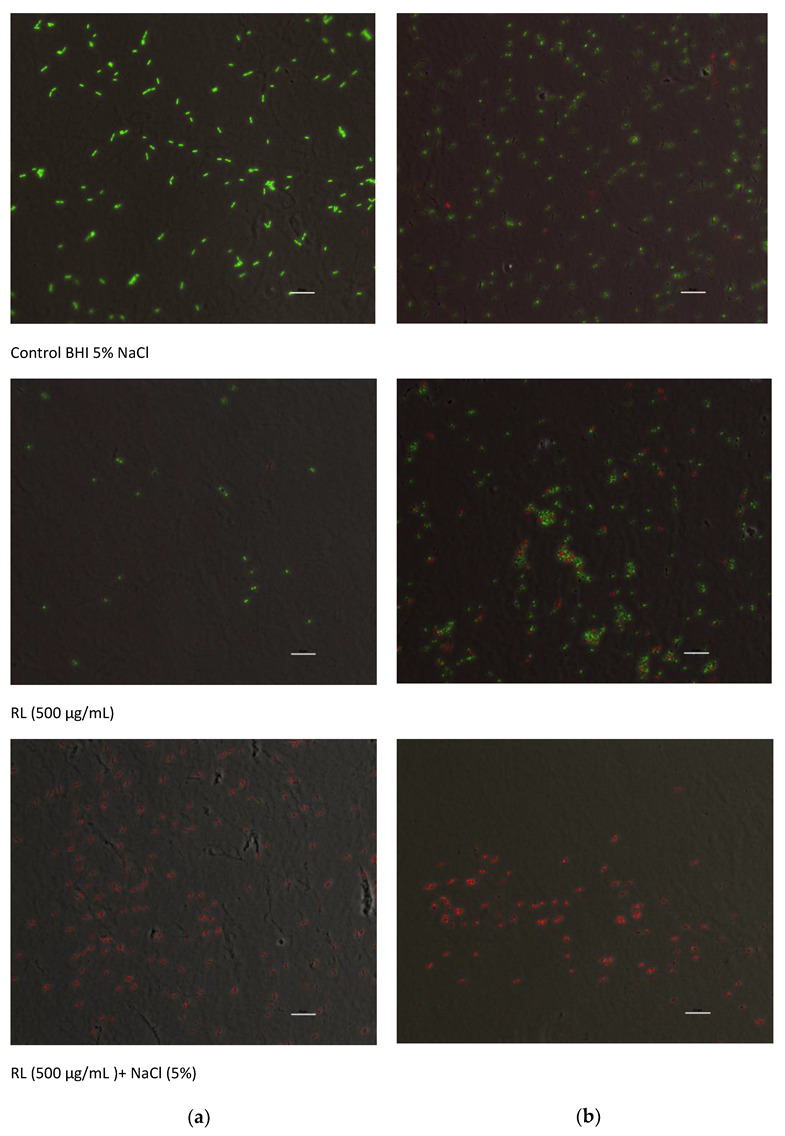
Fluorescence microscopy images of *L. monocytogenes* Scott A after incubation in BHI medium supplemented with NaCl (5%), rhamnolipids (500 µg/mL) and the combination of both for (**a**) 4 h and (**b**) 24 h at 30 °C. Green and red cells correspond to live and dead cells, respectively. Scale bar 10 µm.

**Figure 5 foods-12-00570-f005:**
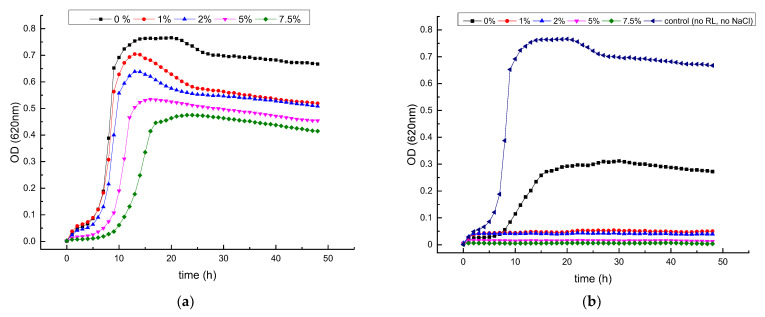
Growth kinetics of *L. monocytogenes* Scott A in BHI medium supplemented with (**a**) different concentrations of KCl, without RL and (**b**) in combination with 500 µg/mL of RL at 30 °C. Data points represent the mean values of three replicates. Error bars have been omitted for clarity.

**Figure 6 foods-12-00570-f006:**
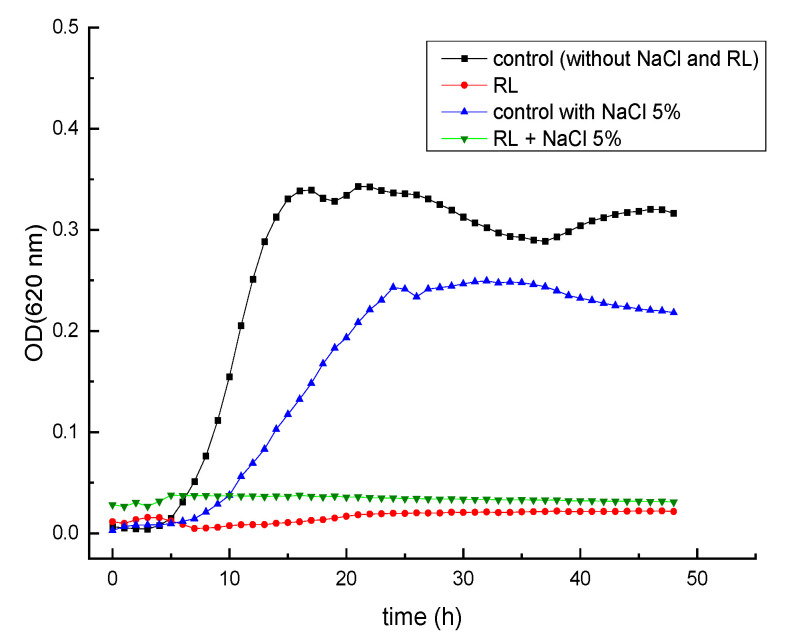
Growth kinetics of *L. monocytogenes* Scott A in BHI medium at pH 5.0 with and without the addition of 500 µg/mL of RL and 5% NaCl at 30 °C. Data points represent the mean values of three replicates. Error bars have been omitted for clarity.

**Table 1 foods-12-00570-t001:** MIC and MBC (between brackets) values for RL (µg/mL) against *L. monocytogenes* Scott A at different NaCl and KCl concentrations.

Salt Concentration	0.0%	1.0%	2.0%	5.0%	7.5%
NaCl	>2500 (-)	2500 (-)	312 (-)	78 (78)	39 (39)
KCl	>2500 (-)	312 (625)	156 (156)	78 (78)	39 (39)

-: no MBC observed.

**Table 2 foods-12-00570-t002:** Critical micelle concentration values of RL solutions under different pH and salt levels.

**%NaCl**	**0.0%**	**1.0% (171 mM)**	**2.0% (342 mM)**	**5.0% (855 mM)**
pH 5.0	19.6	11.5	10.4	11.9
pH 6.0	93.2	32.0	17.4	14.2
pH 7.0	94.2	35.8	26.3	16.0
pH 8.0	121.4	39.1	25.6	15.5
**% KCl**	**0.0%**	**1.0% (134 mM)**	**2.0% (268 mM)**	**5.0% (670 mM)**
pH 7.0	94.2	46.9	38.3	21.7
**% KCl**		**1.3% (171mM)**	**2.5% (342 mM)**	**6.4% (855 mM)**
pH 7.0		25.5	19.5	13.4

## Data Availability

Data is contained within the article.

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
