# Peer review of "Hurdle Technology Approach to Control Listeria monocytogenes Using Rhamnolipid Biosurfactant"

_foods, 2023, doi:10.3390/foods12030570_

Round 1
Reviewer 1 Report
Dear Editors and authors,
A- Major comments:
1-The rhamnolipids compounds, these compounds work on surface tension, but the World Health Organization did not allow them to be used in food, and it is known Listeria monocytogenes bacteria are found in dairy and meat products, how is it added???
B-Minor comments
1-Introduction in the manuscript needs to add some paragraphs that show the presence of Listeria monocytogenes in food in a detailed and clear way. I suggest you to read
1-Yassin, S. A., Nadhom, B. N., & Al-Gburi, N. M. (2021). Detection of Listeria monocytogenes in Several Types of Frozen Meat in Baghdad city. Iraqi Journal of Science, 742-750.
2-Niamah, A. K. (2012). Detection of Listeria monocytogenes bacteria in four types of milk using PCR. Pakistan Journal of Nutrition, 11(12), 1158.
2-The objective of the study is not clear to the reader and needs to be rephrased again in order to clarify the problem to be solved through this study.
3- Many of the work methods mentioned in the manuscript do not contain scientific references, and these methods are vague and incomprehensible. See Growth curves method, Fluorescence microscopy method, .......etc.
4-Page 3 lines 118-121, the authors mentioned a statistical analysis of the work method, while we have a chapter on the statistical analysis in which these details must be mentioned, not with the work method.
5-Figure 1, the y-axis should be renamed to (Log. of bacteria numbers CFU/ ml) .
6-The point 5 applies to Figures 2 .
7-Figure 2 has two columns, these two columns are not defined and unclear.
8-Figure 4 Silent and unclear images. Definitions must be added to clarify the image for the reader.
9-The conclusions in the manuscript are very weak.
Reviewer 2 Report
The manuscript “Hurdle technology approach to control Listeria monocytogenes using rhamnolipid biosurfactant” shows the use of a rhamnolipid to control one of the main pathogens in humans. However, major changes are required.
The authors propose a hurdle technology for the control of L. monocytogenes, low activity of water (glycerol-NaCl), heat, and biosurfactant; however, this technology is only applied for the inactivation of the pathogen cells, while the complementary assays heat factor was omitted.
In lines 59 to 61, they mention that rhamnolipids are not toxic to humans at low concentrations; what are those low concentrations? The page they refer to in the manuscript (https://de3.ai/rhamnolipid.html ) has “safe” in quotes.
Line 88: And the remaining 21.4%, to which molecules does it belong?
Lines 96, how did you determine that at 19 h, L. monocytogenes is in the stationary phase?
What were the percentages of NaCl and glycerol to obtain a 0.92 aw?
Line 105: Why 1.3x109 CFU/mL? What is the reference or standard that supports this?
Line 110: What are those appropriate concentrations of RL? Specify.
Lines 110 to 113: Detailing the inactivation process is unclear and confusing. It must be "RL or 60ºC" or "RL and 60ºC"
Section 2.4, they should not first determine the MIC and MBC to perform the inactivation assays.
Line 134, specify what are the "pre-defined concentrations" of RL.
Line 133, why do they use 17 CFU in this assay and 1.39 CFU in the inactivation test?
Line 131, L. monocytogenes is a facultative anaerobe; why use "an oxygen-impermeable adhesive foil" if previous tests had been carried out under aerobic conditions?
Line 136: Specify why they made the growth curve with KCL from materials and methods.
Section 2.8. Has the CMC of this rhamnolipid not been determined? The concentrations of KCl and NaCl were the same as in section 2.5?
Lines 174 and 210, Error bars represent the standard deviation of the means of triplicates.
Lines 179 and 180, Fernandez et al. 2007, do not conclude that “bacterial heat resistance increases when reducing aw of the treatment medium”; they conclude that “the effect of the aw on the thermal resistance of L. monocytogenes may vary depending upon the temperature of the heat treatment”
Line 189, the concentrations do not correspond to those used to determine the MIC.
Page 7, do not refer to https://doi.org/10.1016/j.foodcont.2012.06.009. They must discuss the results of this study.
Line 269, because with 500 μg/mL, if this concentration was not evaluated in the inactivation.
Line 270, Review, since at 5% NaCl a significant decrease in the growth rate is not observed with respect to lower % (Fig. 3A).
Lines 242 and 243, Review the proposed mechanisms at https://doi.org/10.1016/j.biortech.2020.123206
Table 1, the MIC is not always the same as the MBC; they should review the technique and apply the formula to calculate the MBC. How did you determine the MBC? Here are two references: “Briefly, the MBC were determined as the lowest extract concentration that killed 99% of bacteria in the initial inoculums within 24 h.” https://doi.org/10.1016/j.foodcont.2012.06.009 “Antimicrobials are usually regarded as bactericidal or fungicidal if the MBC/MIC or MFC/MIC ratio is ≤4 and bacteriostatic or fungistatic if > 4” https:/ /doi.org/10.1016/j.archoralbio.2020.104690
Table 2, Please show the CMC results of KCl at pH 5, 6, and 8.
To have continuity in the results, they must perform the growth kinetics of L. monocytogenes with KCl and RL. Trials with KCl should be deleted or completed in the manuscript.
The conclusions are scarce; they need to show general conclusions of the work and perspectives (the last paragraph of the previous section is suggested to be placed in conclusions).
The RL was provided by Rhamnolipid Inc. or for Johnson & Johnson
Reviewer 3 Report
The paper describes a hurdle technology approach to control Listeria monocytogenes using rhamnolipid biosurfactant. This is an interesting topic, and few research papers are available investigating the synergistic effect of rhamnolipid as a bacterial surfactant on the inactivation of L. monocytogenes in food systems. Minor changes must be made in the Introduction section. My main question is how the authors presented the microbiological results in Sections 3.1 and 3.2. I need clarification, but after improving these and adding some more highlights in the conclusion section, the paper can be published in the high-impact factor journal of Foods.
Comments
Line 16. Please, reverse the values, starting from the lowest to the highest. Please, do the same throughout the manuscript (e.g. Lines 20, 189 etc.)
Lines 18 & 20. Please, revise the word “reductions” to “reduction”.
The introduction section includes all the important information and some more recent references need to be added. In I addition I would suggest the authors include some information about the ability of L. monocytogenes to grow in adverse environments with high NaCl, as they are using a high NaCl concentration of up to 7.5%.
Line 34. Please, split the sentence in two and include the reference below:
Ekonomou, S. I., & Boziaris, I. S. (2021). Non-thermal methods for ensuring the microbiological quality and safety of seafood. Applied Sciences, 11(2), 833.
Line 47. Please, include a more recent reference on the use of natural antimicrobial alternatives.
Line 61. I am not sure if you can use a link as a reference, please, check the guide for authors.
Line 64. Please, do not forget to include seafood products in the wide list of L. monocytogenes vehicles.
Line 106. Why the authors are reporting the results as Log CFU/mL? However, in the abstract, the results are presented as log reduction.
Lines 110-111. What was the final population of the treated L. monocytogenes cells in the Eppendorf tubes after inoculation?
Lines 113-115. Please, mention the detection limit of this method, as the drop plate method has a lower sensitivity. In addition, please, clearly mention the volume of the drops.
Section 3.1. I can understand what the authors are trying to state here, but the way they present their data confuses the reader. Why the authors chose this way instead of a figure clearly showing the logarithmic reduction of the cells in Log CFU/mL after each treatment?
Figures 1 & 2. Please, consider changing the results’ presentation according to my comments and do not overlook adding the units on the Y-axis.
Lines 194-195. Please, rephrase the sentence.
Table 1. This Table is unnecessary. Please, remove it and mention the MIC value in the text and that no MBC has been observed. Please, include a reference, if possible, mentioning the MBC value of RL against L. monocytogenes or other microorganisms.
Conclusions. The conclusions section is limited. Please, add some information about the importance of your results for the food industry to help you highlight them!

Round 2
Reviewer 1 Report
Dear Editors,
The authors have made all required corrections. The manuscript is ready for publication. I agree to publish in the current form.
Reviewer 2 Report
I appreciate your responses and corrections to the manuscript.